# Concentrations of Ca, Mg, P, Prostaglandin E2 in Bones and Parathyroid Hormone; 1,25-dihydroxyvitamin D3; 17-β-estradiol; Testosterone and Somatotropin in Plasma of Aging Rats Subjected to Physical Training in Cold Water

**DOI:** 10.3390/biom11050616

**Published:** 2021-04-21

**Authors:** Mateusz Bosiacki, Izabela Gutowska, Katarzyna Piotrowska, Anna Lubkowska

**Affiliations:** 1Department of Functional Diagnostics and Physical Medicine, Pomeranian Medical University in Szczecin, Żołnierska 54 Str., 71-210 Szczecin, Poland; anna.lubkowska@pum.edu.pl; 2Department of Medical Chemistry, Pomeranian Medical University in Szczecin, Powstańców Wlkp. 72 Str., 70-111 Szczecin, Poland; izagut@poczta.onet.pl; 3Department of Physiology, Pomeranian Medical University in Szczecin, Powstańców Wlkp. 72 Str., 70-111 Szczecin, Poland; piot.kata@gmail.com

**Keywords:** calcium (Ca), magnesium (Mg), phosphorus (P), bone density, exercise in cold water, parathyroid hormone (PTH), 1,25-dihydroxyvitamin D3, prostaglandin E2 (PGE2), 17-β estradiol, somatotropin (GH), testosterone (T)

## Abstract

Exposure to low temperatures can be considered a stressor, which when applied for a specific time can lead to adaptive reactions. In our study we hypothesized that cold, when applied to the entire body, may be a factor that positively modifies the aging process of bones by improving the mechanisms related to the body’s mineral balance. Taking the above into account, the aim of the study was to determine the concentration of calcium (Ca), magnesium (Mg), and phosphorus (P) in bones, and to examine bone density and concentrations of the key hormones for bone metabolism, namely parathyroid hormone (PTH), somatotropin (GH), 1,25-dihydroxyvitamin D3, 17-β estradiol, testosterone (T) in plasma, and prostaglandin E2 (PGE2) in the bone of aging rats subjected to physical training in cold water. The animals in the experiment were subjected to a series of swimming sessions for nine weeks. Study group animals (male and female respectively) performed swimming training in cold water at 5 ± 2 °C and in water with thermal comfort temperature (36 ± 2 °C). Control animals were kept in a sedentary condition. Immersion in cold water affects bone mineral metabolism in aging rats by changing the concentration of Ca, Mg, and P in the bone, altering bone mineral density and the concentration of key hormones involved in the regulation of bone mineral metabolism. The effect of cold-water immersion may be gender-dependent. In females, it decreases Ca and Mg content in bones while increasing bone density and 17-β estradiol and 1,25-dihydroxyvitamin D3 levels, and with a longer perspective in aging animals may be positive not only for bone health but also other estrogen-dependent tissues. In males, cold water swimming decreased PTH and PGE2 which resulted in a decrease in phosphorus content in bones (with no effect on bone density), an increase in 1,25-dihydroxyvitamin D3, and increase in T and GH, and may have positive consequences especially in bones and muscle tissue for the prevention of elderly sarcopenia.

## 1. Introduction

Exposure to cold water is a major challenge for the thermoregulatory processes in warm-blooded organisms. Cold is one of the strongest physiological and psychological environmental stressors, resulting in significant physiological defense reactions in the body, i.e., cold shock response. The body also reacts with an adaptive long-term change in metabolic, physiological, biochemical, and hormonal parameters [1,2,3,4,5,6]. It has also been shown that repeated exposure to cold stimulates the immune system [7,8,9], slows down the heart rate, decreases cardiac output, and increases blood pressure. The influence of cold water initiates a series of processes aimed at maintaining a constant temperature inside the body, which results in increased oxygen uptake and consumption, and local narrowing of blood vessels. It has been shown that this peripheral vasoconstriction may result in reduced filtration of fluid from the vessels into the interstitial tissue space, and consequently alleviate inflammation [10,11]. At the same time, the direct effect of exposure to cold water is a reduction in nerve conduction velocity, which results in reductions in the stretching reflex and muscle spasticity, thus reducing pain during intense muscle work [12]. It has also been shown that cold water immersion reduces the concentration of cortisol and increases the concentration of noradrenaline and dopamine [6,12,13,14].

Despite the increasing confidence of doctors and scientists that a “sedentary, overfed and agitated” lifestyle favors the development of civilization diseases (currently the most frequent causes of death), understanding of the importance of these factors for maintaining health is still inadequate. There is also insufficient understanding of the role that simple lifestyle changes can play in preventing these diseases, especially in the elderly. At the same time, due to various technological improvements and “civilizational progress”, people are less and less exposed to the extreme burdens of environmental factors such as low temperatures or intense physical effort. As a consequence, homeostatic compensatory mechanisms are not fully activated, hence significantly reducing their effectiveness, where even low-intensity stressors may overburden the body and become dangerous to health, especially in the elderly and those with reduced physiological adaptation.

One of the key factors in “healthy aging” is exercise and a balanced diet. Physical effort, even bathing in cold water (e.g., in natural water reservoirs), is commonly regarded as having a positive effect on mental and physical health. However, it is supported by little scientific evidence and research on the effects of exposure to cold, so exercise is usually treated as a separate thing. In addition, although literature provides data on the impact of cold and exercise on selected physiological and biochemical blood parameters, there are no data on the effect on mineral metabolism, such as in bones. Therefore, it seems relevant to perform an analysis of the effect of cold-water exercise on the key elements of bone metabolism [15].

Bone mineral metabolism is a lifelong reconstruction. It occurs with the participation of osteoclasts resorbing it and rebuilding osteoblasts. After the matrix is saturated with hydroxyapatite crystals, osteoblasts become osteocytes. The processes of resorption by osteoclasts and the formation by osteoblasts are closely interrelated and are responsible for maintaining bone tissue homeostasis [15]. Proper bone remodeling depends on the supply of the right amount of energy compounds, nutrients, oxygen, and key elements such as calcium, phosphates, magnesium, and amino acids. There is also a significant influence of hormones regulating bone tissue metabolism, such as insulin, cortisol and thyroxine, along with PTH, calcitonin, sex hormones and vitamin D3 metabolites [16,17,18].

Parathyroid hormone (PTH) is a polypeptide hormone synthesized in the main cells of the parathyroid glands and is secreted as an inactive prohormone into the blood during the low level of ionized calcium in the blood. The hormone secretion is stopped when the calcium level rises to 3 mmol/L. PTH restores the normal concentration of calcium in the blood thanks to the direct influence of the hormone on bones and kidneys metabolism. By stimulating the synthesis of 1,25-dihydroxyvitamin D3 in the kidney, PTH also affects the absorption of calcium in the intestine [19]. The action of PTH in the bones increases osteolysis via an increase in the permeability of osteoblast membranes for calcium ions. The increase in intracellular Ca concentration causes the osteoblasts to contract, and the free spaces between these cells are filled by the osteoclast ruffle border, which in turn causes resorption. The effect of PTH on bone tissues is crucial for the proper concentration of calcium ions in the extracellular fluid, which enables the action of calcitriol, i.e., 1,25-dihydroxyvitamin D3, a vitamin D3 derivative which stimulates osteoclast differentiation. As a steroid hormone 1,25-dihydroxyvitamin D3 can enter to the cells and bind to the proteins of the cell nucleus, canceling the repressors’ effect on the genes responsible for encoding calcium-binding proteins involved in the intestinal calcium absorption. 1,25-dihydroxyvitamin D3 has a stimulating effect on the differentiation of osteoclasts and activation of osteolysis, which causes a very high concentration of calcium and phosphate ions in the extracellular fluid and other parts of the cell (i.e., the cytosol and bone cell mitochondria), which enables action PTH [20].

17-β-estradiol, synthesized in the ovaries, is an estrogen with the most pronounced effect on bones. It exerts direct (via receptors) and indirect (via various other hormones, growth factors and cytokines) effects on bone mineral metabolism. Acting via the receptors on the surface of bone cells, estrogen stimulate the secretion of insulin-like growth factor (IGF-1), transforming growth factor beta-1 (TGF β-1), which positively affects osteoblast maturation, collagen secretion and activation of alkaline phosphatase. Estrogens also inhibit the secretion of interleukin-6 (IL-6) and therefore have an anti-resorptive effect [21]. In males the processes of bone development are conducted through aromatization of testosterone (T) to estradiol [16].

Somatotropin (growth hormone, (GH)) is an anabolic hormone released from pituitary gland in pulsatory and sex-dependent manner. It influences bone mass gain during growth and puberty as peak bone mass is achieved during peak GH in serum [17]. GH, indirectly through IGF-1, promotes osteoblasts differentiation, increases their activity and collagen deposition [17]. Mouse models of IGF-1 depletion indicate GH/IGF-1 influence on osteoclasts maturation, fusion and activity (reviewed in [17]).

Prostaglandin E2 (PGE2) is also active in bone remodeling. The factor stimulating the secretion of PGE2 is parathyroid hormone, while estrogens and glucocorticoids have an inhibitory effect. The action of prostaglandin is complex because it inhibits osteoclast activity in the first few minutes, and its prolonged action activates the replication of osteoclast precursors and stimulates their maturation. Exercise is the factor that inhibits the secretion of prostaglandins, inhibits the activity of osteoclasts, and stimulates the activity of osteoblasts [22].

Exposure to low temperatures can be considered a stressor, which when applied for a specific time can lead to adaptive reactions [6,23,24]. Therefore, in our study we hypothesized that cold, when applied to the entire body, may be a factor that positively modifies the aging process of bones by improving the mechanisms related to the body’s mineral balance. Taking the above into account, the aim of the study was to determine the concentration of calcium (Ca), magnesium (Mg) and phosphorus (P) in bones, to examine bone density, and concentrations of the key hormones for bone metabolism: PTH, 1,25-dihydroxyvitamin D3 and 17-β-estradiol, T, GH in plasma and prostaglandin E2 in the bone of aging rats subjected to physical training in cold water.

## 2. Materials and Methods

### 2.1. Animals

Procedures involving animals were carried out in strict accordance with international animal care guidelines, and every effort was made to minimize suffering and the number of animals used. The experiments were approved by the Local Ethical Committee on Animal Testing at the Pomeranian Medical University in Szczecin, Poland (approval No 4/2014).

### 2.2. Design

The study was conducted on 64 Wistar rats of both sexes (32 males and 32 females), aged 15 months at the beginning of the experiment. Before starting the research, the animals were quarantined in the animal facility for a month. Throughout quarantine and during the experiments, the rats were fed with standard LSM (granulated mixture for mice and rats) and received tap water to drink ad libitum. Rats were randomly assigned to the study groups:

Control group (*n* = 16 animals); animals kept in sedentary conditions: Control group male (*n* = 8); Control group female (*n* = 8).

Study group 5 °C (*n* = 24 animals); animals performing swimming training in cold water at 5 ± 2 °C: Group 5 °C males (*n* = 12); Group 5 °C females (*n* = 12).

Study groups 36 °C (*n* = 24 animals); animals subjected to swimming training in water with thermal comfort temperature (36 ± 2 °C): Group 36 °C males (*n* = 12); Group 36 °C females (*n* = 12).

All animals were kept in the animal facility in specialized plastic cages with open work covers, with litter made of solidified wood dust. The animals were bred in pairs in conventional polipropylen cages (ANIMALAB, Poznań, Poland). The animal house had a constant temperature (23 ± 2 °C), constant air humidity (approx. 40%), and a 12-h day and night cycle. Four animals of the same group were housed in a single cage with free access to dry food and water.

### 2.3. Experimental Procedure

The animals in the experiment were subjected to a series of swimming sessions for 9 weeks. Before starting the swimming session, each individual from the test groups was placed for 2 min in an empty glass tank (dimensions: length 100 cm, width 50 cm, depth 50 cm) to accustomise. Acquaint them to contact with the person responsible for the training sessions and the experimental environment. This operation was repeated for the 7 days preceding the beginning of swimming sessions. During the first week of the study, the duration of the first swim was 2 min and increased by 0.5 min per day up to 4 min on the fifth day of the first week. The duration of the single sessions was based on the results of other researchers who showed that rats would swim actively for up to 4 min in 4–5 °C water, and then only drift [25]. Based on literature data, it is known that rats at a thermally comfortable temperature can be active for much longer [26], and hence the choice of a 4-min daily swimming session for both study groups. In this way, the water temperature in the glass tank in which the animals were swimming was the only variable that differed between both study groups.

From the second to the eighth week of the experiment, the swimming time was 4 min per day. The sessions were conducted every day, 5 days a week, in the morning hours, each time ending by 12 a.m. Only one rat was swimming in each glass tank at a time. After each swimming session, each rat was carefully blotted dry with a paper towel and wrapped in dry lignin in a separate cage for a few minutes to prevent excessive chilling of the body, then placed in their home cages. The rats in the control group were also familiarized with the experimenter daily and placed in an empty glass dry tank for 4 min. At the end of the study, 48 h after the last swimming training, the animals were dissected under anesthesia with ketamine hydrochloride/xylazine (100/10 mg/kg body weight administered intraperitoneally. While the animals were dissected, blood was sampled from the hearth of all animals, and then plasma was obtained (heparinized vacutainer blood collection tubes (Sarstedt, Poznań, Poland) were used). The femurs of the rats were removed and immediately placed in liquid nitrogen. The samples were then stored at –80 °C until further determinations of element concentrations and densitometric analysis were performed.

### 2.4. Determination of the Concentration of Calcium, Magnesium and Phosphorus in Bones by Inductively Coupled Plasma Atomic Emission Spectrometry (ICP-OES)

Samples were analyzed using an inductively coupled plasma optical emission spectrometer (ICP-OES), (ICAP 7400 Duo, Thermo Scientific, Warsaw, Poland) equipped with a concentric nebulizer and cyclonic spray chamber, to determine the Ca, Mg, P content. Analysis was performed in radial and axial modes. The samples were thawed at room temperature and dried overnight at 80 °C to a constant weight after cleaning of all adherent tissue. The bones were ground into powder in a porcelain mortar and mineralized using CEM Corporation MARS 5 Digestion Microwave System (LabX, Midland, ON, Canada). The weight of the bone sample was at least 0.1 g. The samples were transferred to clean polypropylene tubes. Then, 5 mL of 65% HNO_3_ (Suprapur, Merck, Poznań, Poland) was added to each vial and each sample was allowed 30 min pre-reaction time in the clean hood, then 1 mL of non-stabilized 30% H_2_O_2_ solution (Suprapur, Merck, Poznań, Poland) was added to each vial. Once the addition of all reagents was complete, the samples were placed in special Teflon vessels and heated in the microwaved digestion system for 35 min at 180 °C (15 min ramp to 180 °C and maintained at 180 °C for 20 min). At the end of digestion, all samples were removed from the microwave and allowed to cool to room temperature. In the clean hood, samples were transferred to acid-washed 15 mL polypropylene sample tubes. A further 80-fold dilution was performed prior to ICP-OES measurement. A sample of 125 µL was taken from each digest. The samples were spiked with an internal standard to provide a final concentration of 0.5 mg/L Ytrium, 1 mL of 1% Triton (Triton X-100, Sigma, St. Louis, MO, USA), and diluted to a final volume of 10 mL with 0.075% nitric acid (Suprapur, Merck, Poznań, Poland). Samples were stored in a monitored refrigerator at a nominal temperature of 4 °C until analysis. Blank samples were prepared by adding concentrated nitric acid (80 µL) to tubes without sample and subsequently diluted in the same manner described above. ICP multi-element standard solution IV: Ca, Mn, Zn, Cu, Fe, Na, Pb, Cr, Co, Sr, P, Mg (Merck, Darmstadt, Germany) was prepared with different concentrations of inorganic elements in the same manner as the blanks and samples. Deionized water (Direct Q UV, Millipore, approximately 18.0 MΩ) was used for preparation of all solutions. Samples of reference material (NIST SRM 1486 Bone Meal) were prepared in the same manner as the samples. The wavelengths used: Ca 315.887 nm, P 178.284 nm, Mg 280.270 nm. 

The procedures were validated with certified reference material, namely NIST 8414 Bovine Muscle (National Institute of Standards and Technology). The concentration values of the reference materials given by the manufacturers and our determinations are shown in Table 1.

### 2.5. Analysis of Bone Density—Densitometry

Densitometric analysis of the femurs from the rats determined bone mineral density (BMD, g/cm^2^) using the Hologic Horizon DEXA System^®^, (Quirugil, Bogotá, Columbia, Discovery software, version 12.3, Bellingham, WA, USA) with additional software for small animals. The test consisted of scanning the analyzed area with dual energy X-ray absorptiometry using a high and low energy beam. During the measurement, some of the radiation is absorbed and some scattered. The intensity of the radiation at the detector depends on the bone thickness and mineral content. The result of the measurement is the absolute value of the mineral density, expressed in grams of mass (bone mineral content—BMC). Dividing this number by the measurement area gives the bone mineral density in g/cm^2^ (BMD). The measuring device was calibrated before each series of measurements.

### 2.6. Concentration Analysis PGE2 Levels in Bone

The bone samples were treated using a previously published procedures [4,5]. Femoral bone marrow was obtained by flushing the shafts with 4 °C ethyl acetate. Values were normalized with respect to bone weight. The samples were centrifuged at 4 °C, 2000× *g* for 30 min, and the supernatant was collected. The ethyl acetate was evaporated using a stream of nitrogen gas. The residue was resuspended in enzyme immunosorbent assay buffer (EIA) provided by the kit manufacturer, and stored at −80 °C. The PGE2 analysis used a commercially available EIA kit according to the manufacturer’s instructions (Cayman Chemical Co, Ann Arbor, MI, USA; Cat. No. 514010). The sensitivities for PGE2: 15 pg/mg protein.

### 2.7. Protein Assay

Total protein concentration in the bone was measured using a MicroBCA Protein Assay Kit (Thermo Scientific, Pierce Biotechnology, Waltham, MA, USA) and a spectrophotometer (UVM340, ASYS, Dornstadt, Germany), [27].

### 2.8. Plasma Hormone Analysis

The levels of studied hormones were measured using a commercially available enzyme-linked immunosorbent assay (ELISA) in accordance with the manufacturer’s instructions. For PTH: Rat PTH (Parathyroid hormone) ELISA Kit ER0388, Fine Test (BIOKOM, Janki, Poland); For 1,25-dihydroxyvitamin D3: ELISA Kit CEA467Ge, CloudClone (Poznań, Poland); For 17-β-estradiol: ELISA Kit AB 10 8667, Abcam (Poznań, Poland); For testosterone: ELISA Kit MBS282195, MyBioSource (Poznań, Poland); For somatotropin: ELISA Kit MBS019990, MyBioSource (Poznań, Poland). The sensitivities: parathormon—7.5 pg/mL; 1,25-dihydroxyvitamin D3—4.1; 17-β estradiol—8.68 pg/mL; testosterone—pg/mL, 0.04 ng/mL; somatotropin—0.1 ng/mL.

### 2.9. Histological Analysis

Femoral bones cleared from muscle tissue were collected and fixed in formalin (4% Chemland, Stargard, Poland) for 24 h, then decalcified in 10% EDTA solution (Merck, Poznań, Poland). After decalcification, the soft bones were embedded in paraffin blocks and cut into 3 μm slides.

The deparaffinised sections of decalcified bone (3 μm) were rehydrated in xylene and decreasing concentrations of ethanol (99.8–50%) then transferred to deionized water. After rehydration, the sections were stained with Mayer’s haematoxylin and eosin (H&E) according to standard procedures. After staining, the sections were dehydrated in 95% and 99.8% alcohol, cleaned with xylene, mounted with Canada balsam (all from Sigma-Aldrich, Poznań, Poland), and observed under an Olympus IX81 inverted microscope (Olympus, Warsaw, Poland). Micrographs were collected using CellSens software (Olympus, Warsaw, Poland). Bone morphology on decalcified long bone sections was examined. The thickness of the collagen layer in cortical bone in each experimental group was measured on bone slides (one section from each experimental animal of every experimental group; and 6 measures on each slide) using Cell Sense software (“measure arbitrary line” tool).

### 2.10. Statistical Analysis

Analysis was performed using STATISTICA v12.5 PL (StatSoft Polska). The normality of the distribution of the results of the evaluated parameters was tested using a Shapiro-Wilk test, and descriptive statistics (mean, standard deviation, median, upper and lower quartile, minimum and maximum) were determined. A Mann–Whitney U test was performed to compare the significance of differences between the studied groups, which were deemed significant at *p* < 0.05.

## 3. Results

### 3.1. Concentration of Calcium in the Bones of the Rats

Bone Ca in the female rats swimming in cold water (5 °C) was significantly lower than in the controls (by approx. 6%, *p* = 0.002). Bone Ca was significantly lower in the female rats who swam in cold water than those in the warm water group (by approx. 3%, *p* = 0.026). A statistically significantly lower bone Ca concentration was found in the warm water group than in the cold-water group (by about 10%, *p* = 0.04) (Figure 1).

### 3.2. Concentration of Magnesium in the Bones of the Rats

Bone Mg in the group of female rats swimming in cold water (5 °C) was significantly lower than in the controls (by approx. 5%, *p* = 0.002). Bone Mg in the cold-water group was significantly lower than in the warm water group (by approx. 5%, *p* = 0.03). Bone Mg in the male rats swimming in cold water (5 °C) was significantly lower than in the controls (by approx. 33%, *p* = 0.002) and also significantly lower in the warm water group (36 °C) than in the controls (by approx. 39%, *p* = 0.04). Bone Mg was significantly lower in the warm water group than in the cold-water group (by approx. 8%, *p* = 0.01) (Figure 2).

### 3.3. Concentration of Phosphorus in the Bones of the Rats

Bone P in the females swimming in warm water (36 °C) was statistically significantly higher than in the cold-water group (5 °C group) (by approx. 3%, *p* = 0.03). Bone P in the males swimming in warm water (36 °C) was statistically significantly lower than both the controls (by approx. 15%, *p* = 0.002) and the cold-water group (by approx. 8%, *p* = 0.002) (Figure 3).

### 3.4. Bone Mineral Density of the Rats

Bone mineral density of the female rats swimming in cold water (5 °C) was statistically significantly higher than in the controls (by 12%, *p* = 0.03), (Figure 4).

### 3.5. Plasma Parathyroid Hormone in the Female and Male Rats

Plasma PTH in the female rats swimming in cold water (5 °C) was statistically significantly lower than in the controls (by about 54%, *p* = 0.002), similar to those swimming in warm water (36 °C) (by 46%, *p* = 0.002). Plasma PTH was also significantly lower in the cold-water group than in the warm water group (by about 18%, *p* = 0.022). Plasma PTH in the male rats swimming in cold water (5 °C) was statistically significantly lower than in the controls (by 33%, *p* = 0.002), similar to those swimming in warm water (36 °C) (about 20% lower than in the controls, *p* = 0.0025). The male rats swimming in cold water had statistically significantly lower plasma PTH than in the warm water group (by about 20%, *p* = 0.002), (Table 2 and Table 3).

### 3.6. Plasma 1,25-dihydroxyvitamin in the Female and Male Rats

Plasma 1,25-dihydroxyvitamin in the female rats swimming in cold water (5 °C) was statistically significantly higher than in the controls (about 5%, *p* = 0.042), as in the warm water group (36 °C) (about 5% higher than in the controls, *p* = 0.022). In the cold-water group, plasma 1,25-dihydroxyvitamin was statistically significantly higher than in the warm water group (by about 7%, *p* = 0.022). Plasma 1,25-Dihydroxyvitamin in the male rats swimming in cold water (5 °C) was statistically significantly higher than in the controls (by 16%, *p* = 0.022) as in the warm water group (36 °C) (by about 7%, *p* = 0.042). In the cold-water group, plasma 1,25-dihydroxyvitamin was statistically significantly higher than in the warm water group (by about 8%, *p* = 0.022) (Table 2 and Table 3).

### 3.7. Plasma 17-β Estradiol in the Female Rats

Plasma 17-β estradiol in the female rats swimming in cold water (5 °C) was statistically significantly higher than in the controls (by about 25%, *p* = 0.03) as in the warm water group (36 °C) (by about 13%, *p* = 0.02). Plasma 17-β estradiol in the female rats swimming in cold water was statistically significantly higher than in the warm water group (by about 8%, *p* = 0.02) (Table 2).

### 3.8. Plasma Testosterone in the Male Rats

Plasma testosterone in the male rats swimming in cold water (5 °C) was statistically significantly higher than in the controls (by 13%, *p* = 0.021) as well as in the warm water group (36 °C) (by about 7%, *p* = 0.046). In the cold-water group, plasma testosterone was statistically significantly higher than in the warm water group (by approximately 5%, *p* = 0.021) (Table 3).

### 3.9. Plasma Somatotropin in the Male and Female Rats

Plasma somatotropin in the female rats swimming in cold water (5 °C) was statistically significantly higher than in the controls (by about 28%, *p* = 0.015) as in the warm water group (36 °C) (by about 19%, *p* = 0.032). Plasma somatotropin was statistically significantly higher in the cold-water group than in the warm water group (by about 7%, *p* = 0.022). Plasma somatotropin in the male rats swimming in cold water (5 °C) was higher than in the controls (by 63%, *p* = 0.042) as in the warm water group (36 °C) (by about 55%, *p* = 0.022). In the cold-water group, it was higher than in the warm water group (by about 7.5%, *p* = 0.020) (Table 2 and Table 3).

### 3.10. Bone Prostaglandin E2 in the Female and Male Rats

Plasma PGE2 in the female rats swimming in cold water (5 °C) was significantly lower than in the controls (by about 15%, *p* = 0.003) as in the warm water group (36 °C) (by 5%, *p* = 0.02). In the cold-water group, plasma PGE2 was lower than in the warm water group (by about 12%, *p* = 0.025). Plasma PGE2 in the male rats swimming in cold water (5 °C) was lower than in the controls (by 33%, *p* = 0.002) as in the warm water group (36 °C) (by about 20%, *p* = 0.0025). In the cold-water group, plasma PGE2 was lower than in the warm water group (by about 17%, *p* = 0.002) (Table 2 and Table 3).

### 3.11. Histological Analysis

The Figure 5 show representative images of the bone tissue sections. In our study a proper structure of articular cartilage and trabecular bone were observed (not shown). Examination of compact bone of the rats in all the experimental groups revealed a parallel arrangement of collagen type I fibers and the presence of osteocytes in lacunae (Figure 5, asterisk). The thickness of the cortical layer of the long bones was greater in the rats swimming in cold water (5 °C) in both sexes (by about 25% in females, *p* = 0.002 and by 15% in males, *p* = 0.001 appropriately) and without change in the rats swimming in warm water, in comparison to the controls.

## 4. Discussion

This study of the effect of 9 weeks of daily swimming in cold water (5 °C) and thermally comfortable water (36 °C) on bone mineral metabolism of aging rats, showed a significant effect of cold-water immersion on bone tissue, expressed as changed Ca, Mg and P levels in the bone with a concomitant increase in bone mineral density. The cold-water swimming also had effects on plasma PTH, 1,25-dihydroxyvitamin, 17-β estradiol, T, GH and PGE2 concentrations in the bones of the rats. These effects were stronger in animals exercising in cold water than in the warm water, where changes in the concentrations of the studied elements were observed, namely a significant decrease in P concentration in the bones of the male rats swimming in warm water and a statistically insignificant decrease in Ca and Mg in the bones of the female rats.

### 4.1. The Effect of Gender

The observed effect of swimming in cold water on the bone differed between the sexes of the rats. In the females, it resulted in a decrease in Ca, Mg and P content in the bones while increasing bone mineral density. In the males on the other hand, the cold-water swimming decreased the Mg content in the bones without any effect on bone mineral density. In the literature to date, there are no data on the effect of cold on bone metabolism, therefore it is difficult to relate our results to the results of other authors. The same model of study had been used previously [12,28], where resulted in body weight change in animals. Male rats swimming in low temperatures showed a decrease in body weight, which, as suggested by the authors of the study, was associated with mobilization of adipose tissue storage and increase energy expenditure. In contrast, when swimming at a thermally comfortable temperature, the decrease in body weight in males and females was due solely to the exercise. The authors also noted that the females that took part in swimming sessions at 5 °C at first had a decrease in body weight in the first two weeks of the experiment, then an increase until the end of sessions Since the present study is a continuation of that research and was performed on available tissues from rats from the cited experiment, it may provide valuable data for the analysis of the effect of cold-water immersion on bone mineral metabolism. It seems that the increase in bone density and body weight in females with a simultaneous decrease in concentrations of the elements studied, may be indicative of intensive bone remodeling and increased synthesis of organic bone constituents, which in the longer term should lead to an increase in Ca, Mg, and P contents in the bones as a result of the initiation of mineralization of this newly synthesized bone tissue. It takes about 10 days from the moment of osteoid formation to the beginning of mineralization. Osteocytes in the first phase accumulate Ca and P, which are then secreted outside and combine with collagen. The process of matrix mineralization begins with the deposition of amorphous calcium phosphate, which is then transformed into a crystalline form containing phosphates, calcium, carbonates and citrates with the participation of magnesium ions [15]. In the male rats swimming in cold water, a reduction in body weight was observed [29,30], with no effect on bone mineral density or Ca and P concentrations, but a reduction in Mg content. Again, this may indicate the initiation of bone remodeling processes, but at a slower rate than in females. It seems that a large role in this process may be played by the greater muscle mass of the males compared to the females, which then may participate in tremor thermogenesis, thus ensuring thermal homeostasis. At the same time, the muscular work is accompanied by an increased demand for Mg, which could be the reason for its “mobilization” and the observed decrease in its content in the bones. Therefore, it seems reasonable to conclude that cold may be a factor that positively modifies bone aging by improving the mechanisms that influence bone density.

### 4.2. The Role of Ca, Mg, P in Bone Mineral Metabolism

The changes in Ca, Mg, and P concentrations observed in the present study in the bones of rats swimming daily in cold water may be of significant importance for the process of bone mineralization. It is important for this process not only to have sufficient concentrations of elements, but also to have the right proportions. The interaction between Ca and Mg is crucial, as these elements have different functions in bone tissue. Calcium is the basic building block of bone, and its optimal supply influences the correct growth and mineralization of bone tissue in early development, allowing peak bone mass during adolescence, and to maintain it in adult life. Proper Ca supply also delays the development of physiological osteopenia and reduces the risk of osteoporosis in post-menopausal age and later [15,29]. From the total amount of calcium (1000 g), up to 99% is deposited in the bone tissue in the form of hydroxyapatite, while 1% of the bone pool (as a fast-exchange pool) can be exchanged with calcium in the extracellular fluid [15,30,31].

The most important role of Mg for bone tissue metabolism is the activation of alkaline and acid phosphatase, pyrophosphatase and ATP-ase. Mg also stabilizes amorphous calcium phosphate and inhibits the formation of crystalline hydroxyapatite [32,33,34]. With an adequate amount of magnesium in the bone, it is also possible to remodel apatites low in calcium ions (with a Ca/P ratio of 1.3–1.5) from an amorphous and immature form to a crystalline and mature form (hydroxyapatite with a Ca/P ratio of more than 1.66) [33,35]. An incorrect Ca/Mg ratio can cause bone demineralization (osteoporosis). Hypomagnesemia leads to hypocalcemia despite a sufficient Ca^2+^ intake and normal urinary excretion. However, in a study in rats, no significant changes in bone were observed under low magnesium diet conditions [33].

The present study examined only bone mineral density and not its physical properties, such as hardness and resistance to torsion, and does not discuss any such potential abnormalities in the bones of the studied rats resulting from swimming in cold water. However, the histological analysis revealed a proper structure of the cortical bone, with a parallel arrangement of collagen I fibers, and osteocytes in lacunae in all experimental groups. Measurements of the thickness of the cortical bone performed on the slides in our study showed increased thickness in the rats that were swimming (hot and cold water, both genders), with a big increase in the female rats swimming in cold water, which confirms data from the bone mineral density analysis.

Two weeks of the daily swimming sessions, especially in cold water (stress conditions), caused an increase in secretion of anabolic hormones: GH in both sexes, testosterone in males, and estradiol in females, which are known to influence collagen I production and deposition in cartilage and bone tissue [36,37]. The increased sex hormones and GH were already described for different exercise protocols and there is an interaction between GH and T release during exercise and in the recovery period [38,39]. There is also a correlation between the estradiol level and amount of GH released in the recovery period, and GH secretion is blunted in ovariectomised (low estradiol) rats [40].

### 4.3. The Role of Low Temperature and Hormonal Regulation in Bone Mineral Metabolism

A low temperature combined with physical exercise results in increased production of energy, which is released in the form of heat, accompanied by increased oxygen uptake and consumption, and increased production of reactive oxygen species (ROS). Increased synthesis of ROS may result in oxidative stress and cellular damage, such as in lipid peroxidation. In a previously cited study by Lubkowska et al. (2019) [28] conducted on the same rat model, a significant increase of lipid peroxidation markers (such as thiobarbituric acid reactive substances (TBARS) and 8-isoprostane) in erythrocytes was demonstrated in the rats swimming in either cold water or thermally comfortable water, as well as in the control group. The increase in ROS synthesis may result in increased prostanoids synthesis through activation of cyclooxygenases (COX). There are two isoforms of COX enzymes: COX-1 which is constitutively expressed in most cells, and COX-2 which is induced by inflammation and contributes to the production of PGE2 [41]. PGE2 is a very active local modulator of bone remodeling, which in the first few minutes after its release inhibits osteoclast activity, and over a longer term exerts a stimulating effect on osteoclast formation and maturation [22]. It is known that physical activity can induce the production of piezoelectric currents in bone, increase collagen production, and inhibit the production of PGE2. This may explain the changes in mineral metabolism and bone density observed in the present study. The concentration of PGE2 in the bones of both the female and male rats swimming in cold water was lower than in the controls. In addition to temperature, bone metabolism and remodeling may also have been modified by hydrostatic pressure. Studies confirm that dynamic hydrostatic pressure has a positive influence on bone development and is an important stimulus by which osteochondral cells and their progenitors sense and respond to mechanical loading. It was found that the mineralized portion of the developing femur cultured under any cyclic hydrostatic pressure regime were significantly larger and/or denser than the non-stimulated controls [42].

The increased bone density may have also been associated with an increased concentration of 1,25-dihydroxyvitamin D3 observed in our study. Similar results were obtained in a study conducted in elderly Japanese men, where it was shown that 5 weeks of endurance training could inhibit the seasonal reduction in serum 1,25-dihydroxyvitamin D3 concentrations without changes in body fat [43]. The same authors in another study on the effects of an acute bout of endurance exercise on vitamin D concentrations showed that the increase in 1,25-dihydroxyvitamin D3 concentration was greater in men than in women [44]. The increase in 1,25-dihydroxyvitamin and PTH concentrations observed in our study may promote renal tubule reabsorption of calcium and inhibit the reabsorption of phosphorus, thereby reducing urinary calcium excretion and increasing urinary phosphate excretion [45]. The interaction with PTH and 1,25-Dihydroxyvitamin D3 ensures the maintenance of calcium homeostasis. In addition, adequate Ca and P concentrations in the extracellular fluid in the periosteal tissue are essential for normal bone mineralization. Although the bone appears to be a stable structure, there is a dynamic balance between the process of bone mineralization and the process of Ca and P release from the bone, leading to an increase in plasma Ca and P. Any decrease in plasma Ca concentration immediately increases the synthesis of PTH, which is a stimulator of 1,25-dihydroxyvitamin D3 synthesis in the kidney. Both hormones promote Ca resorption from bone. Consequently, increased plasma 1,25-dihydroxyvitamin D3 concentrations contribute to a decrease in plasma PTH. Thus, this is a precise mechanism that is important for the proper functioning of the body.

Summarizing, immersion in cold water affects bone mineral metabolism in aging rats by changing the concentration of calcium, magnesium and phosphorus in the bone, altering bone mineral density and altering the concentration of key hormones in the regulation of bone mineral metabolism. The effect of cold-water immersion was gender-dependent. In females it decreases Ca and Mg content in bones while increasing bone density and 17β-estradiol and 1,25-dihydroxyvitamin D3 levels, what in longer perspective in aging animals may be positive not only for bone health but also other estrogen-dependent tissues. In males, cold water swimming decreased PTH and PGE2 which resulted in a decrease in phosphorus content in bones (with no effect on bone density), an increase in 1,25-dihydroxyvitamin D3, and increase in T and GH, which also may have positive consequences especially in bones and muscle tissue for the prevention of elderly sarcopenia.

## 5. Conclusions

Based on the results and discussion of this study, we may conclude that cold can be a factor that positively modifies the aging process of bones by improving the mechanisms that affect their density. The effect of cold-water immersion was gender-dependent.

### Limitation of the Study and Perspectives for Future Research

In this experiment, transverse cross-sections of long bones were not performed, which consequently did not allow for histological determination of the formation of secondary osteons. Therefore, bone remodeling could only be determined by the thickening of the collagen layer of compact bone visible in the longitudinal section, and indirectly by the amount of mineral composites.

The study does not indicate whether the positive effect of cold-water training on bone ageing processes, through an improvement in the body’s mineral balance, would persist into old age. Therefore, it would be interesting to conduct further research in this area.

## Figures and Tables

**Figure 1 biomolecules-11-00616-f001:**
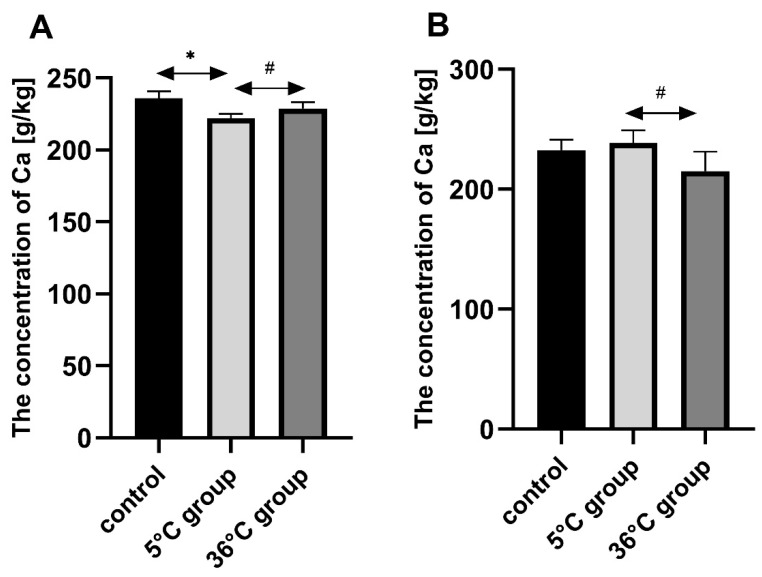
Bone Ca [g/kg] in female (**A**) and male (**B**) rats from the control group and the test groups. Rats were subjected to 4 min swimming sessions for 9 weeks in either cold water at 5 °C or at a thermally comfortable temperature of 36 °C. The first swimming session was 2 min long (day one) increasing by 0.5 min per day to 4 min (day five of the first week). Results are presented as means and standard deviations. * *p* < 0.05 the level of statistical significance vs. the control group (Mann–Whitney U test). ^#^
*p* < 0.05 level of statistical significance vs. the 5 °C group (Mann–Whitney U test).

**Figure 2 biomolecules-11-00616-f002:**
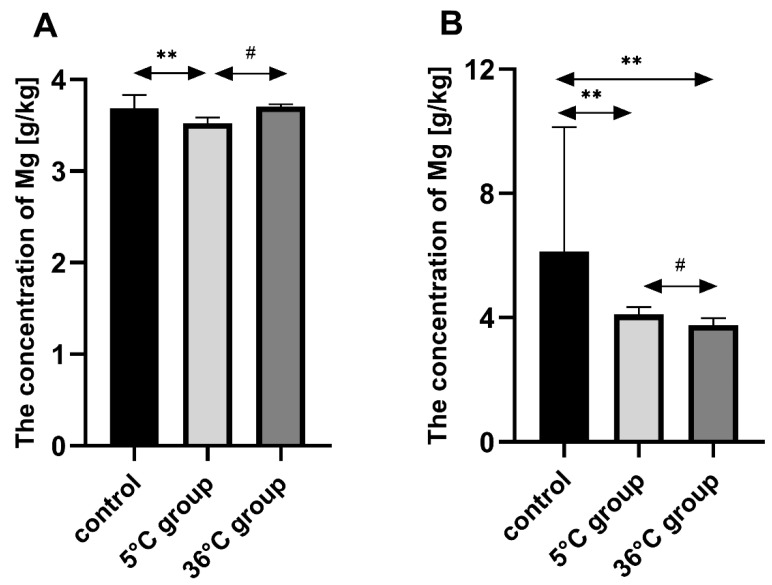
Bone Mg [g/kg] in female (**A**) and male (**B**) rats from the control group and the test groups. Rats were subjected to 4 min swimming sessions for 9 weeks in either cold water at 5 °C or at a thermally comfortable temperature of 36 °C. The first swimming session was 2 min long (day one) increasing by 0.5 min per day to 4 min (day five of the first week). Results are presented as means and standard deviations. ** *p* < 0.005 level of statistical significance vs. the 5 °C group (Mann–Whitney U test). ^#^
*p* < 0.05 level of statistical significance vs. the 5 °C group (Mann–Whitney U test).

**Figure 3 biomolecules-11-00616-f003:**
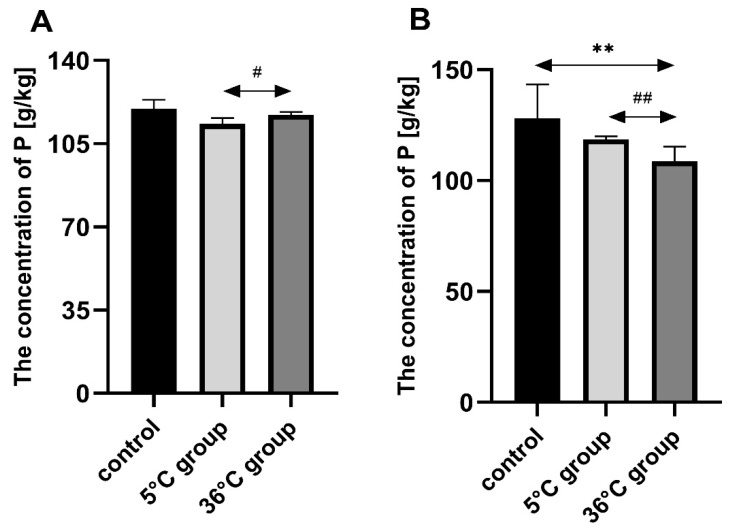
Bone P [g/kg] in female (**A**) and male (**B**) rats from the control group and the test groups. Rats were subjected to 4 min swimming sessions for 9 weeks in either cold water at 5 °C or at a thermally comfortable temperature of 36 °C. The first swimming session was 2 min long (day one) increasing by 0.5 min per day to 4 min (day five of the first week). The results are presented as means and standard deviations. ** *p* < 0.005 level of statistical significance vs. the control group (Mann–Whitney U test). ^#^
*p* < 0.05 level of statistical significance vs. the 5 °C group (Mann–Whitney U test). ^##^
*p* < 0.005 level of statistical significance vs. the 5 °C group (Mann–Whitney U test).

**Figure 4 biomolecules-11-00616-f004:**
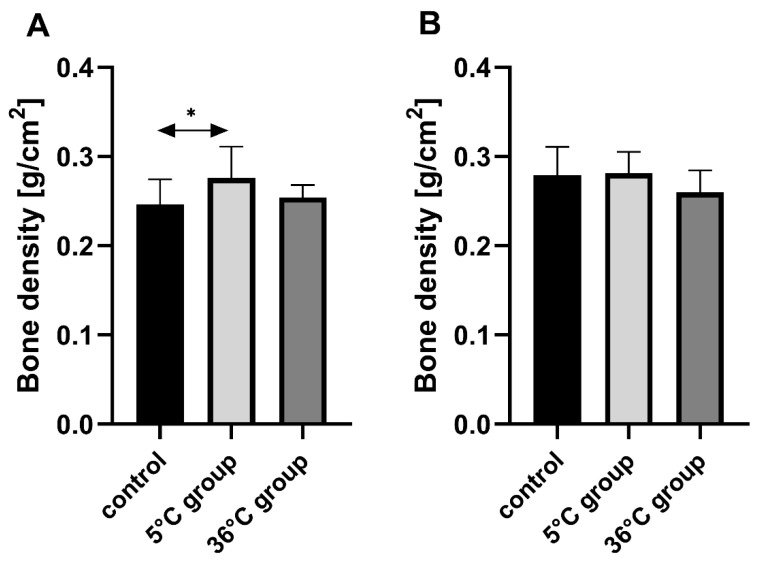
Bone mineral density [g/cm^2^] in the bones of female (**A**) and male (**B**) rats in the control and test groups. Rats were subjected to 4 min swimming sessions for 9 weeks in either cold water at 5 °C or at a thermally comfortable temperature of 36 °C. The first swimming session was 2 min long (day one) increasing by 0.5 min per day to 4 min (day five of the first week). The results are presented as means and standard deviations. * *p* < 0.05 level of statistical significance vs. the control group (Mann–Whitney U test).

**Figure 5 biomolecules-11-00616-f005:**
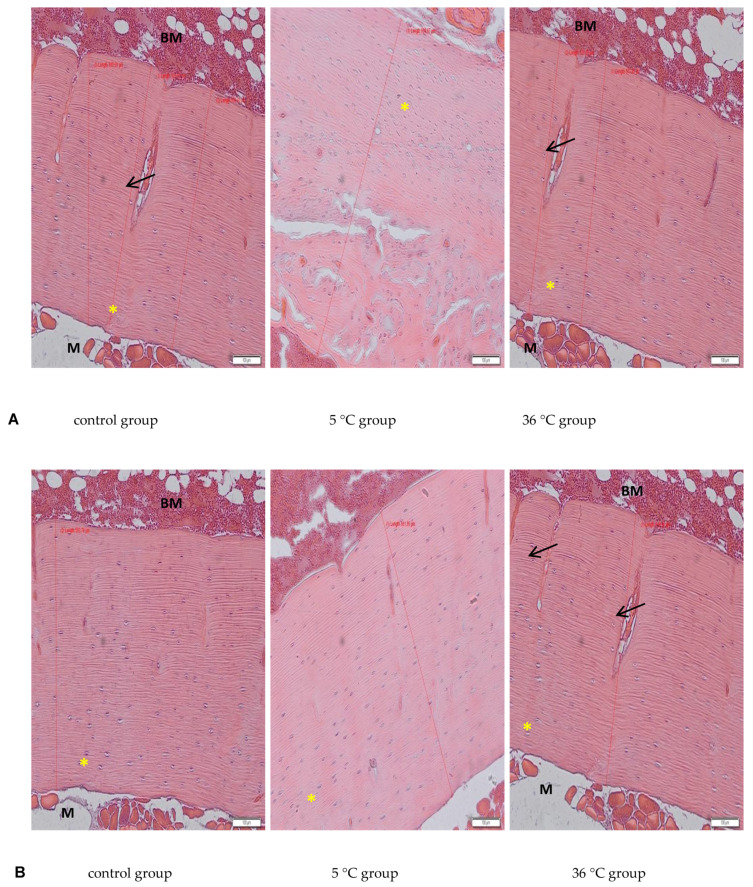
Histological analysis of the bones of female (**A**) and male (**B**) rats from control and test groups. Rats were subjected to 4 min swimming sessions for 9 weeks in either cold water at 5 °C or at a thermally comfortable temperature of 36 °C. The first swimming session was 2 min long (day one) increasing by 0.5 min per day to 4 min (day five of the first week). Examination revealed a thicker layer of cortical bone in the rats swimming in cold water. HE staining, M—skeletal muscle tissue, BM—bone marrow; yellow asterisk—osteocytes in lacunae, black arrow—transverse channels in bone with blood vessels; ×10 objective magnification, original scale bar 100 μm, diameters of cortical bone measured in Olympus microscope software CellSense (Olympus, Warsaw, Poland), showed in μm scale. Only representative images are presented.

**Table 1 biomolecules-11-00616-t001:** Table analysis of NIST-SRM 1486 (Bone Meal) [g/kg].

Bone Meal SRM NIST 1486			
Chemical Elements	Certified	Measured *n* = 6	Recovery (%)
Mg	4.600	4.365	94.900
Ca	265.800	251.032	94.440
P	123.000	131.500	106.910

**Table 2 biomolecules-11-00616-t002:** Plasma levels of parathyroid hormone (PTH); 1,25-dihydroxy-vitamin D3; 17-β estradiol, somatotropin in plasma, and bone levels of prostaglandin E2 (PGE2) in female rats from the control and test groups.

Biochemical Parameters	PTH [pg/mL]	1,25-dihydroxy-vitamin D3 [ng/mL]	17-β Estradiol [pg/mL]	Somatotropin [pg/mL]	PGE2 [pg/mg Protein]
	Plasma	Plasma	Plasma	Plasma	Bone
	Females				
control	66.48 ± 5.38	25.67 ± 12.09	20.05 ± 6.15	796.40 ± 44.62	48.05 ± 2.65
5 °C group	30.59 ± 4.50 **	29.05 ± 6.16 *	25.09 ± 4.28 *	1021.80 ± 56.94 *	40.96 ± 5.34 *
36 °C group	35.96 ± 12.88 **^#^	27.01 ± 5.62 *^#^	23.09 ± 6.15 *^#^	944.01 ± 42.50 *^#^	45.76 ± 3.02 *^#^

Rats were subjected to 4 min swimming sessions for 9 weeks in either cold water at 5 °C or at a thermally comfortable temperature of 36 °C. The first swimming session was 2 min long (day one) increasing by 0.5 min per day to 4 min (day five of the first week). Results are presented as means±standard deviations. * *p* < 0.05 level of statistical significance vs. the control group (Mann–Whitney U test). ** *p* < 0.005 level of statistical significance vs. the control group (Mann–Whitney U test). # *p* < 0.05 level of statistical significance against the 5 °C group (Mann–Whitney U test).

**Table 3 biomolecules-11-00616-t003:** Plasma levels of parathyroid hormone (PTH); 1,25-dihydroxy-vitamin D3; testosterone and somatotropin in plasma, and bone levels of prostaglandin E2 (PGE2) in male rats from the control and test groups.

Biochemical Parameters	PTH [pg/mL]	1,25-dihydroxy-vitamin D3 [ng/mL]	Testosterone [pg/mL]	Somatotropin [pg/mL]	PGE2 [pg/mg Protein]
	Plasma	Plasma	Plasma	Plasma	Bone
	Males				
control	63.09 ± 15.4	28.23 ± 12.95	46.02 ± 4.31	675.44 ± 50.13	51.03 ± 2.35
5 °C group	42.27 ± 12.90 **	32.87 ± 3.05 *	52.03 ± 2.35 *	1106.85 ± 35.67 *	46.02 ± 4.31 *
36 °C group	50.78 ± 16.22 *^#^	30.15 ± 1.35 *^#^	49.26 ± 3.18 *^#^	1049.32 ± 66.93 *	48.26 ± 3.18 *^#^

Rats were subjected to 4 min swimming sessions for 9 weeks in either cold water at 5 °C or at a thermally comfortable temperature of 36 °C. The first swimming session was 2 min long (day one) increasing by 0.5 min per day to 4 min (day five of the first week). Results are presented as means±standard deviations. * *p* < 0.05 level of statistical significance vs. the control group (Mann–Whitney U test). ** *p* < 0.005 level of statistical significance against the control group (Mann–Whitney U test). ^#^
*p* < 0.05 level of statistical significance vs. the 5 °C group (Mann–Whitney U test).

## Data Availability

Not applicable.

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
