# Peer review of "Concentrations of Ca, Mg, P, Prostaglandin E2 in Bones and Parathyroid Hormone; 1,25-dihydroxyvitamin D3; 17-β-estradiol; Testosterone and Somatotropin in Plasma of Aging Rats Subjected to Physical Training in Cold Water"

_biomolecules, 2021, doi:10.3390/biom11050616_

Round 1
Reviewer 1 Report
Comments and suggestions are in the attached document.

Author Response
Review 1
Thank you very much for a kind review, all comments were carefully corrected in accordance with the reviewer's remarks.
Abstract:
Line 17-22 Testosterone determination not mentioned
We sincerely apologize for this oversight, according to Reviewer remark we corrected the sentence
„During the first week of the study, the duration of the first swim was 2 minutes and increased by 0.5 minutes per day up to 4 minutes on the fifth day of the first week.“-not relevant in the abstract, but please ad data concerning different water temperature (groups)
According to Reviewer remark we corrected the Abstract and ad data concerning different water temperature and groups
Line 33 The abbreviations GH and T were not previously defined
According to Reviewer remark we corrected theabbreviations
Line 36 Omit cold therapy since exercise in cold water is not considered a form of cold therapy that is more used as a superficial thermal modality
According to Reviewer remark we corrected the key words
Introduction:
Line 53 ...nerve conduction velocity
According to Reviewer remark we corrected the sentence
Line 79 Just 15% osteoblast differentiate into osteocytes, those entrapped into the bone matrix, please rewrite
This line and entire paragraph in our manuscript is not describing the fate of osteoblasts and generation of osteocytes. We don’t know how to respond to this comment.
Line 81 Citation not in Vancouver style citation
According to Reviewer remark we corrected the citation
Line 113 Citation not in Vancouver style citation
According to Reviewer remark we corrected the citation
Please add the main hypothesis of the study.
According to Reviewer remark we added hypothesis of the study
M&M
Line 185-190 It is not clear how the euthanasia was performed. With a 10 mg/kg IM of ketamin is hardly possible to maintain a deep state of anaesthesia. Please follow AVMA Guidelines for the Euthanasia of Animals: 2020 Edition
How were plasma samples collected?
We sincerely apologize for this oversight, according to Reviewer remark we corrected the description of euthanasia:
“At the end of the study, 48 hours after the last swimming training, the animals were dissected under anesthesia with ketamine hydrochloride/xylazine (100/10 mg/kg body weight administered intraperitoneally. While the animals were dissected, blood was sampled from the hearth of all animals, and then plasma was obtained (heparinized vacutainer blood collection tubes (Sarstedt, Poznań, Poland) were used)”.
Line 226 The table should be the first not third.
Thank you very much for this Reviewer remark, however Tab. 3 belongs to the material and methods section and shows the content of tested elements in the certified bone material, not in our samples. Therefore, in our opinion, it cannot be transferred to the results, because it concerns the method. We would appreciate the possibility of leaving Table 3 in Materials and methods section.
Line 248 What do you mean by protein concentration in the bone. Which proteins? There is no data about it in the result section. Please explain.
We apologize for the oversight. Total protein determination was performed to calculate the concentration of PGE2 in the bones. This is a standard expression of the tissue concentration of a test compound.
Line 262 Was the evaluator aware of the samples group?
Thank you very much for this attention. Of course, all the trials were coded and the evaluator did not know what trials he was analysing.
Results
Line 428-435 Please avoid several sentences with "we" as a first word.
According to Reviewer remark we corrected the sentence
There are no data about the cortical thickness, it is not clear if any statistical analysis was used to evaluate differences between groups. Please explain.
According to the Reviewers suggestion we rewritten the point concerning bone thickness estimation
From all figures and tables omit „The first swimming session was 2 minutes long (day one) increasing by 0.5 minutes per day to 4 minutes day five of the first week.“ Please change the colour of one of the groups since the two grey colours are very similar.
According to Reviewer remark we corrected all figures
Please consider making a single figure merging all bone minerals and bone mineral density since it could make the results easier to follow.
Thank you very much for this remark of the reviewer, however, we would prefer to keep the division of figures as they are now, because combining them would also make it necessary to reduce their size, and then the histological picture would be completely unreadable
The size of the tables is not appropriately matched to the page size.
Thank you very much for this remark of the reviewer, however tables have been formatted according to authors' guide
Line 388-396 Cursive unnecessary
According to Reviewer remark we corrected the sentence
Figure 5. is not giving any additional value to the results so I recommend omitting it.
Thank you very much for this comment from the reviewer, however, we would prefer to keep Figure 5, which shows the histological analysis of the bones examined. Other reviewers also expressed a positive opinion about the changes in the bones we observed, which is why we would like to leave the photos taken by us with such an effort
Please add statistical analysis on cortical thickness.
According to the Reviewers suggestion we added statistical analysis.
Discussion
How would you explain an increase in bone mineral density noted in female rats with a simultaneous decrease in Ca, Mg and P content in bone?
According to Reviewer remark weincluded an explanation in the discussion section:
“It seems that the increase in bone density and body weight in females with a simultaneous decrease in concentrations of the elements studied, may be indicative of intensive bone remodeling and increased synthesis of organic bone constituents, which in the longer term should lead to an increase in Ca, Mg and P content in the bones as a result of the initiation of mineralization of this newly synthesized bone tissue. It takes about 10 days from the moment of osteoid formation to the beginning of mineralization. Osteocytes in the first phase accumulate Ca and P, which are then secreted outside and combine with collagen. The process of matrix mineralization begins with the deposition of amorphous calcium phosphate, which is then transformed into a crystalline form containing phosphates, calcium, carbonates and citrates with the participation of magnesium ions [15]”.
Line 465 Reference number 12 is not published in 2019, changed with 25?
According to Reviewer remark we corrected the reference
Line 524-527 Please add data in the M&M and Results section (see previous comments).
According to Reviewer remark we corrected the Material and Methods and Results section
Line 569 There was a decrease not an increase in PTH in the present study, please rewrite.
According to Reviewer remark we corrected the sentence
Study limitations missing, please add.
According to Reviewer remark we added study limitations

Reviewer 2 Report
See atached PDF file

Author Response
Review 2
Minor points concerning my view on the Manuscript ID biomolecules-1148206
Please insert a blank space before % symbol and also insert blanks before and after
= symbol.
Check the font style (italics?) in point 3.6
Please insert a blank space before oC units in Figure 5.
In ‘5. Conclusions, Based on the results of our study, it can be concluded’... Pleasewrite:
‘5. Conclusion, Based on the results and discussion of this study, we be conclude’…
Thank you very much for a kind review, all comments were carefully corrected in accordance with the reviewer's remarks.

Reviewer 3 Report
The present study examined the concentration of Ca, Mg, P in femoral bones and levels of the key hormones for bone metabolism in plasma and bone of aging rats subjected to physical training in cold water. The study is well designated and includes all relevant references. However, the results are not clearly presented, subsequently discussed and some methods should be more precise described. I have several suggestions and recommendations which should lead to an improvement of the manuscript.
Abstract
- all abbreviations provided should be explained in the text before
- it is not necessary to provide methods (e.g. ICP-OES, ELISA) on the basis of which the monitored parameters were determined
- according to your results, the effect of cold water immersion was gender-dependent. Therefore, the sentence “The effect of cold water immersion may be gender-dependent” should be edited.
Introduction
- the citations of [Clarke 2008 – line 81], [Tringali 2014 – line 113] should be correctly stated
- throughout the manuscript, all abbreviations provided should be explained in the text before
Materials and Methods
- I suggest to divide female rats into the following groups: C1 (control), A1 (group 5 °C), B1 (group 36 °C); male rats into the following groups: C2 (control), A2 (group 5 °C), B2 (group 36 °C) and subsequently to use this designation throughout the manuscript, mainly to clearer present the results obtained.
- how were the animals bred (individually, in pairs)? Complete the cage manufacturer, please.
- the citation of Zhang and Wang 2010 [45] (lines 239-240) should be correctly stated
- correct “-80oC” (line 244)
- the phrase “femur bones” (line 263) should be replaced by “femoral bones”
- the sentence “The figures show representative images of the bone tissue sections” (line 274) should be a part of the Results and not of the Material and Methods
Results
- use proposed designation of groups studied to clearer present your results, please
- when statistically insignificant changes in some parameters were observed between groups, it means there are no significant differences between these groups with regard to those parameters, e.g. “Bone mineral density of the female rats swimming in warm water (36 °C) was higher than in the controls (by approx. 3%) but it was not statistically significant (p=0.23). In the females swimming in cold water, bone mineral density was 8% higher than in the warm water group, but not statistically significantly (p=0.034).” (lines 341-345) – it means there is no significant difference in bone mineral density between C1 and B1 groups as well as A1 and B1 groups - correct all the results given in this way, please
- descriptions of Tables 1, 2 and Figure 5 should be marked in a smaller font
- the results of 3.6. (Plasma 1,25-dihydroxyvitamin in the female and male rats) should not be italicized
- following sentence should be corrected: “Plasma testosterone the male rats swimming in cold water (5 °C) was statistically 405 significantly higher than in the controls (by 13%, p=0.021) as in the warm water group (36 406 °C) (by about 7%, p=0.046)” (lines 405-407)
- following two sentences within the histological analysis (“We examined bone morphology on decalcified long bone sections (line 428)”; “We also measured the thickness of the collagen layer in each experimental group using CellSense software ("measure line" tool)” – lines 432-433) should be a part of the Material and Methods. Please, describe more precise how the thickness of the collagen was measured. How many measurements were performed in histological sections of one individual?
- cancellous bone is not compact bone!; correct the phrase “cancellous (compact) bone” (line 430)
- I would expect that osteons (primary and/or secondary) will also be detected at least in the groups A1, A2. Did you not identify these structures in any group?
Discussion
- it is stated that swimming in cold water is associated with intensive bone remodeling in both female and male rats. Secondary osteons are the result of the process of bone remodeling and they can be observed in the compact bone microstructure of rodents, including rats. Therefore, the discussion in this direction would also be appropriate.
- the citation of Lubkowska et al (2019) [25] (line 541) should be correctly stated
- according to your results, the effect of cold water immersion was gender-dependent. Therefore, the sentence “The effect of cold water immersion may be gender-dependent” (lines 586-587) should be edited.
Conclusion
- it should also include the information related to gender-dependent effect
I suggest the manuscript to be accepted after a major revision.
Author Response
Review 3
The present study examined the concentration of Ca, Mg, P in femoral bones and levels of the key hormones for bone metabolism in plasma and bone of aging rats subjected to physical training in cold water. The study is well designated and includes all relevant references. However, the results are not clearly presented, subsequently discussed and some methods should be more precise described. I have several suggestions and recommendations which should lead to an improvement of the manuscript.
Abstract
- all abbreviations provided should be explained in the text before
We carefully corrected abbreviations in accordance with the reviewer's remarks.
- it is not necessary to provide methods (e.g. ICP-OES, ELISA) on the basis of which the monitored parameters were determined
We removed methods from abstract in accordance with the reviewer's remarks.
- according to your results, the effect of cold water immersion was gender-dependent. Therefore, the sentence “The effect of cold water immersion may be gender-dependent” should be edited.
According to Reviewer remark we added the sentence “The effect of cold water immersion may be gender-dependent” to conclusion.
Introduction
- the citations of [Clarke 2008 – line 81], [Tringali 2014 – line 113] should be correctly stated
According to Reviewer remark we corrected the citation.
- throughout the manuscript, all abbreviations provided should be explained in the text before
According to Reviewer remark we corrected all abbreviationsthroughout the manuscript.
Materials and Methods
- I suggest to divide female rats into the following groups: C1 (control), A1 (group 5 °C), B1 (group 36 °C); male rats into the following groups: C2 (control), A2 (group 5 °C), B2 (group 36 °C) and subsequently to use this designation throughout the manuscript, mainly to clearer present the results obtained.
We are very grateful for Reviewer remark, however we would like to keep the group names used in the manuscript due to our previous manuscripts based on the same research model, which makes it possible to compare the results of our research.
- how were the animals bred (individually, in pairs)? Complete the cage manufacturer, please.
According to Reviewer remark we added information about animals bred and cages
- the citation of Zhang and Wang 2010 [45] (lines 239-240) should be correctly stated
According to Reviewer remark we corrected the citation
- correct “-80oC” (line 244)
According to Reviewer remark we corrected the sentence
- the phrase “femur bones” (line 263) should be replaced by “femoral bones”
According to Reviewer remark we corrected the sentence
- the sentence “The figures show representative images of the bone tissue sections” (line 274) should be a part of the Results and not of the Material and Methods
According to Reviewer remark we removed the sentence from the Material and Methods section and added to the Results section
Results
- use proposed designation of groups studied to clearer present your results, please
- when statistically insignificant changes in some parameters were observed between groups, it means there are no significant differences between these groups with regard to those parameters, e.g. “Bone mineral density of the female rats swimming in warm water (36 °C) was higher than in the controls (by approx. 3%) but it was not statistically significant (p=0.23). In the females swimming in cold water, bone mineral density was 8% higher than in the warm water group, but not statistically significantly (p=0.034).” (lines 341-345) – it means there is no significant difference in bone mineral density between C1 and B1 groups as well as A1 and B1 groups - correct all the results given in this way, please
According to Reviewer remark we removed allinsignificant sentence from the Results section.
- descriptions of Tables 1, 2 and Figure 5 should be marked in a smaller font
According to Reviewer remark we corrected fonts in Tables1-2 and Fig.5
- the results of 3.6. (Plasma 1,25-dihydroxyvitamin in the female and male rats) should not be italicized
According to Reviewer remark we corrected the Result section
- following sentence should be corrected: “Plasma testosterone the male rats swimming in cold water (5 °C) was statistically 405 significantly higher than in the controls (by 13%, p=0.021) as in the warm water group (36 406 °C) (by about 7%, p=0.046)” (lines 405-407)
According to Reviewer remark we corrected the sentence
- following two sentences within the histological analysis (“We examined bone morphology on decalcified long bone sections (line 428)”; “We also measured the thickness of the collagen layer in each experimental group using CellSense software ("measure line" tool)” – lines 432-433) should be a part of the Material and Methods. Please, describe more precise how the thickness of the collagen was measured. How many measurements were performed in histological sections of one individual?
According to the Reviewer suggestion we moved description of measurement method to method section and rephrased that to be more detailed: “The thickness of the collagen layer in cortical bone in each experimental group was measured on bone slides (one section from each experimental animal of every experimental group; and 6 measures on each slide) using Cell Sense software ("measure arbitrary line" tool).”
- cancellous bone is not compact bone!; correct the phrase “cancellous (compact) bone” (line 430)
According to the reviewer suggestion we left phrase:” compact bone”.
- I would expect that osteons (primary and/or secondary) will also be detected at least in the groups A1, A2. Did you not identify these structures in any group?
The osteocytes in the lacunae are pointed with * (asterisks ) on every presented image and in the main text we placed phrase: “Examination of compact bone of the rats in all the experimental groups revealed a parallel arrangement of collagen type I fibers and the presence of osteocytes in lacunae (Figure 5, asterisk).” The secondary osteones can be detected on transverse sections of bones in our study we used longitudinal sections, so we didn’t speculated about presence and amount of secondary osteons.
Discussion
- it is stated that swimming in cold water is associated with intensive bone remodeling in both female and male rats. Secondary osteons are the result of the process of bone remodeling and they can be observed in the compact bone microstructure of rodents, including rats. Therefore, the discussion in this direction would also be appropriate.
We thank the Reviewer for this comment, however, the description and precise localization of primary and secondary osteons can be properly investigated on transverse sections of bones; we obtained longitudinal sections of bones, that is why we didn’t want to speculate about amount of primary/secondary osteons, and we focused on mineralization and bone thickness.
- the citation of Lubkowska et al (2019) [25] (line 541) should be correctly stated
According to Reviewer remark we corrected the citation
- according to your results, the effect of cold water immersion was gender-dependent. Therefore, the sentence “The effect of cold water immersion may be gender-dependent” (lines 586-587) should be edited.
According to Reviewer remark we corrected the sentence
Conclusion
- it should also include the information related to gender-dependent effect
According to Reviewer remark we corrected the conclusion
I suggest the manuscript to be accepted after a major revision.
Thank you very much for a kind review, all comments were carefully corrected in accordance with the reviewer's remarks.

Round 2
Reviewer 3 Report
The authors accepted most of my recommendations and the quality of their manuscript was improved. However, there are still some grammatical errors and/or typos in the text (e.g. fig. 5 show – page 10; egzamined – page 18). According to the response from the authors, they examined longitudinal sections of femoral bones which do not allow the identification of osteons (primary and/or secondary) in compact bone microstructure. Therefore, this fact should be listed in the “Limitation of the study”. It is a pity that the authors did not investigate transversal sections of femoral bones which allow a more detailed examination of compact bone microstructure.
Author Response
Review 3
Comments and Suggestions for Authors
The authors accepted most of my recommendations and the quality of their manuscript was improved. However, there are still some grammatical errors and/or typos in the text (e.g. fig. 5 show – page 10; egzamined – page 18). According to the response from the authors, they examined longitudinal sections of femoral bones which do not allow the identification of osteons (primary and/or secondary) in compact bone microstructure. Therefore, this fact should be listed in the “Limitation of the study”. It is a pity that the authors did not investigate transversal sections of femoral bones which allow a more detailed examination of compact bone microstructure.
Thank you very much for a kind review, all comments were carefully corrected in accordance with the reviewer's remarks.
